# On the testing of grain shape corrections to bedload transport equations with grain-resolved numerical simulations

Yulan Chen<sup>1</sup>, Orencio Durán<sup>2</sup>, and Thomas Pähtz<sup>1</sup>

**Correspondence:** Thomas Pähtz (0012136@zju.edu.cn)

Abstract. Using grain-resolved LES-DEM simulations, Zhang et al. (J. Geophys. Res. Earth Surf. 130, e2024JF007937, 2025) aimed to validate a grain-shape-corrected bedload transport equation proposed earlier by the same group. It states that grain shape effects are captured through a modified Shields number that depends, among others, on the drag coefficient,  $C_{D_{\text{settle}}}$ , determined from the force balance for a grain settling in a fluid at rest. To independently vary  $C_{D_{\text{settle}}}$  in their simulations, the authors changed the boundary conditions on the grains' surfaces: By artificially shifting the locations of the no-slip conditions from the actual grain surface to a virtual surface a distance l into the grain interior, they hoped to well approximate Navier-slip conditions with a slip length l. Here, we argue that this approximation is appropriate only if the thickness of the boundary layer that forms around the virtual surface is much larger than l, which we demonstrate was not the case for the authors' simulations. In particular, using independent DNS-DEM grain settling simulations for the same hydrodynamic conditions, we directly show that this approximation substantially overestimates the value of  $C_{D_{\text{settle}}}$  of a Navier-slip sphere. This implies that the conditions created with their artificial method do not correspond to physically realistic scenarios and therefore do not support the authors' grain shape correction. To support this conclusion, we demonstrate that their entire numerical data can be alternatively explained by a simple null hypothesis model, without grain shape correction, based on the virtual-grain rather than the actual-grain size.

#### 15 1 Introduction

Sediment transport occurs when a sufficiently strong flow of fluid shears a bed of loose sedimentary grains (Pähtz et al., 2020). In the case that the fluid is a liquid and the characteristic volume-equivalent grain diameter  $d_p$  on the order of 1 mm or larger, most grains roll, slide, and hop along and in close vicinity to the bed surface, a regime known as bedload transport (Ancey, 2020a, b). A key interest of many researchers has been the rate q at which bedload transport occurs, here termed bedload flux, when driven by a nearly steady, uniform flow along a nearly flat bed (Ancey, 2020a). Over the last century, numerous equations predicting q have been proposed for such idealized conditions (e.g., Meyer-Peter and Müller, 1948; Bagnold, 1956, 1973; Pähtz and Durán, 2020; Deal et al., 2023). However, while most such studies treated transported grains as spheres, only a single study, to our knowledge, attempted to account for the typically non-spherical shape of transported grains, the one by Deal et al. (2023).

<sup>&</sup>lt;sup>1</sup>Institute of Port, Coastal and Offshore Engineering, Ocean College, Zhejiang University, 316021 Zhoushan, China

<sup>&</sup>lt;sup>2</sup>Department of Ocean Engineering, Texas A&M University, College Station, Texas 77843-3136, USA

They proposed the following bedload flux equation:

25 
$$\frac{q}{d_p \sqrt{(\rho_s/\rho_f - 1)gd_p}} = \alpha_o \left(\frac{C^*}{\mu^*} \frac{\tau_b}{(\rho_s - \rho_f)gd_p} - \tau_{co}^*\right)^{3/2},$$
 (1)

where  $\rho_s$  and  $\rho_f$  are the sediment and fluid densities, respectively, g is the magnitude of the bed-normal component of the gravitational acceleration  $\mathbf{g} = g_x \hat{\mathbf{x}} + g_z \hat{\mathbf{z}} = (Sg, 0, -g)$  (with S the bed slope),  $\tau_b$  the bed shear stress, and  $\alpha_o$  and  $\tau_{co}^*$  are dimensionless constants. Equation (1) resembles the classical bedload flux equation by Meyer-Peter and Müller (1948), but with a Shields number that has been multiplied with the dimensionless coefficient  $C^*/\mu^*$ , where

30 
$$\mu^* \equiv \frac{\mu_s - S}{\mu_o - S}$$
 and  $C^* \equiv \frac{S_f C_{D_{\text{settle}}}}{C_o}$ .

In these expressions,  $\mu_s$  is the static bulk friction coefficient of the granular material,  $\mu_o = \tan(24^\circ)$  its associated value for an assembly of spheres,  $S_f$  the Corey shape factor,  $C_{D_{\text{settle}}}$  the drag coefficient for the settling of a single grain in a fluid at rest (S=0), and  $C_o$  its associated spherical-grain value, calculated from the model by Dietrich (1982) as described by Deal et al. (2023). The settling drag coefficient  $C_{D_{\text{settle}}}$  is determined through the classical balance between the drag, gravitational, and buoyancy forces acting on a grain settling with terminal velocity  $\omega_s$  (Bagnold, 1956):

$$\frac{1}{8}\pi d_p^2 \rho_f C_{D_{\text{settle}}} \omega_s^2 = \frac{1}{6}\pi d_p^3 (\rho_s - \rho_f) g,$$
(3)

resulting in (Deal et al., 2023)

45

50

$$C_{D_{\text{settle}}} = \frac{4(\rho_s/\rho_f - 1)gd_p}{3\omega_s^2}.$$
(4)

The bedload transport equation (1) rests on shaky foundations, since the grain-shape-parametrizing coefficient  $C^*/\mu^*$  varied by less than 20%, only between 0.84 and 1.05, in the experiments by Deal et al. (2023), with four of their five tested grain materials even exhibiting nearly no variation at all  $(C^*/\mu^* \in [1.01, 1.05])$ . Furthermore, they were unable to vary  $C^*$  independently from  $\mu^*$ , which is particularly problematic because the inclusion of  $C^*$  in their correction of the Shields number conflicts with long-standing established knowledge about the physics of sediment transport. In fact, a very large number of successful aeolian and fluvial sediment transport models (Pähtz and Durán, 2018, and references therein), and bedload transport models in particular, are based on Bagnold's hypothesis that the friction coefficient at the interface between sediment bed and transport layer is a sole property of the granular bulk material (Bagnold, 1956). For equilibrium transport conditions, this interface friction coefficient is equal to the ratio between the average streamwise drag and bed-normal submerged gravitational forces acting on transported grains (Bagnold, 1956). Hence, if Bagnold's hypothesis is true, and numerical simulations based on the Discrete Element Method (DEM) suggest that it is (Pähtz and Durán, 2018), then drag-induced effects on transported grains should be insensitive to the qualitative and quantitative nature of the drag force law and, thus, to  $C^*$ .

To address some of these shortcomings, Zhang et al. (2025) conducted grain-resolved bedload transport simulations using Large Eddy Simulation (LES) for the fluid phase coupled with the DEM for the sediment phase consisting of naturally-shaped grains. Then, keeping the grain shape, and thus  $\mu^*$ , constant, they aimed to solely vary  $C^*$  through conducting further simulations with changed boundary conditions at the grains' surfaces: By artificially shifting the locations of the no-slip conditions

from the actual grain surface to a virtual surface a distance l into the grain interior, they hoped to well approximate Navier-slip conditions with a slip length l. Navier-slip conditions, where the tangential component of the slip velocity,  $u_t$ , satisfies  $u_t = l\partial u_t/\partial n$ , are typically used for hydrophobic particles (Tao et al., 2023) or for fluid-particle systems in which the particle size is comparable to the mean-free path ( $\sim l$ ) or characteristic separation distance between fluid molecules, such as for rarefied gases (Tao et al., 2017). In other words, they correspond to physically meaningful scenarios and therefore represent a valid means to numerically probe the phase space of grain properties in the context of bedload transport. However, the same cannot necessarily be said about the, in their own words, "artificial-shrinkage method" that Zhang et al. (2025) used to approximate Navier-slip conditions. In fact, if this approximation method were inappropriate, there would be no good physical justification to base the drag force on the grain size  $d_p$ , as done in equations (3) and (4). Instead, from taking their artificial-shrinkage method literally, one would actually have to base it on the shrunk grain size  $d'_p$  corresponding to the virtual grain surface seen by the LES solver. However, this results in an alternative settling drag coefficient  $C'_{D_{\text{settle}}}$  that is different from  $C_{D_{\text{settle}}}$ :

$$\frac{1}{8}\pi d_p'^2 \rho_f C_{D_{\text{settle}}}' \omega_s^2 = \frac{1}{6}\pi d_p^3 (\rho_s - \rho_f) g, \tag{5}$$

$$\Rightarrow C'_{D_{\text{settle}}} = \frac{4(\rho_s/\rho_f - 1)gd_p^3}{3\omega_s^2 d_p'^2}.$$
 (6)

Note that, in equation (5), the buoyancy force, even though it is also a fluid force, is still calculated based on the actual grain size  $d_p$  rather than  $d_p'$ , since Zhang et al. (2025) computed the buoyancy force by adding  $-\frac{1}{6}\pi d_p^3 \rho_f g_z \hat{z}$  manually to the vertical force on the grains whilst eliminating the actual buoyancy force contribution to the total fluid force via considering only a streamwise driving  $\rho_f g_x \hat{x}$ , but no vertical driving  $\rho_f g_z \hat{z}$ , in the fluid momentum balance.

Here, we show in two distinct manners that the artificial-shrinkage method by Zhang et al. (2025) constitutes, indeed, an inappropriate approximation of Navier-slip boundary conditions for their studied hydrodynamic conditions (§2). First, we analytically estimate that the thickness  $\delta$  of the boundary layer that forms around a settling sphere is, depending on the location on the sphere's surface, comparable or even substantially smaller than the slip length l, even though  $\delta$  would need to be much larger than l for the approximation to make physical sense. Second, using independent Direct Numerical Simulation (DNS)-DEM simulations of a settling sphere, a direct comparison between the values of  $C_{D_{\rm settle}}$  obtained from simulations with Navier-slip conditions and those obtained from simulations based on the artificial-shrinkage method reveals that the former are substantially smaller than the latter. The consequence of these findings is that the conditions Zhang et al. (2025) created with their method do not correspond to physically realistic scenarios and therefore do not support the grain shape correction in equation (1). To support this conclusion, we demonstrate that their entire numerical data can be alternatively explained by a simple null hypothesis model, without grain shape correction, based on the virtual-grain rather than the actual-grain size (§3).

# 2 Zhang et al.'s Navier-slip approximation

Zhang et al. (2025) simulated systems consisting of naturally-shaped grains, each of which created via gluing a number of spheres together. To approximate Navier-slip boundary conditions in their numerical model, they Taylor-expanded the tangential slip velocity  $u_t$  one would expect in the case of Navier-slip conditions,  $u_t = l\partial u_t/\partial n$ , from the surface of each such

100

composite sphere to the surface of a virtual shrunk sphere of a radius that is a distance  $l = \text{Sk}\Delta x$  smaller than the actual radius, where Sk is a shrinkage coefficient and  $\Delta x = 0.5$  mm the grid size of their numerical mesh. Due to  $u_t = l\partial u_t/\partial n$ , the first-order Taylor-expanded value of  $u_t$  at each virtual shrunk composite sphere is then equal to precisely zero, like for a noslip condition. They argued that this simple mathematical result justified approximating Navier-slip conditions on the surfaces of their naturally-shaped grains by no-slip conditions at the corresponding shrunk composite spheres' surfaces. Furthermore, Zhang et al. (2025) claimed that this "artificial-shrinkage method" constitutes "a typical approximation", citing a number of previous studies throughout their paper (Nguyen and Ladd, 2002; Boutt et al., 2011; Cui et al., 2012; Fukumoto et al., 2021; Jiang et al., 2022). However, in actuality, none of these studies were discussing Navier-slip conditions at all. Instead, the study by Nguyen and Ladd (2002) was about lubrication force implementation, while the other cited studies proposed grain shrinkage as a means to artificially match the pore space connectivity of two-dimensional to three-dimensional simulations. In addition, in our own literature research, we were unable to find a single study backing this claim.

Zhang et al. (2025) also presented numerical justification for using their artificial-shrinkage method in their supplementary material (their text S4 and figures S11 and S12). However, the description of the numerical setup underlying their supporting figures S11 and S12 is very vague (e.g., quantitative details of the simulated setup and conditions are completely missing), and their publicly available code does not contain the procedures or modules required to reproduce the simulations behind these figures. Moreover, even after repeated inquiries over a period of several months, Zhang et al. (2025) have remained unwilling to share with us any of the code they used to produce their figures S11 and S12.

In what follows, we present analytical (§2.1) and numerical (§2.2) falsifications of the claim that their artificial-shrinkage method approximates Navier-slip conditions.

#### 2.1 Analytical falsification

In order for the first-order Taylor expansion of the tangential slip velocity  $u_t$  to be a physically reasonable approximation of the fluid-particle velocity difference around a Navier-slip grain's surface, the distance from the surface at which this expansion is evaluated, and therefore the slip length l, must be sufficiently small. In the present case, "sufficiently" means much smaller than the thickness  $\delta(x_s)$  of the boundary layer that forms around the corresponding virtual shrunk no-slip grain, which varies with the location  $x_s$  on its surface, since a distance  $\delta$  away from  $x_s$  in the normal direction, the flow velocity has approximately reached that of the outer layer and therefore no longer conveys any information about the flow disturbance caused by the no-slip boundary conditions. The largest values of  $\delta$  are expected to occur at surface locations  $x_s$  where the flow separates. For an order-of-magnitude estimate of  $\delta$  at such points, let us consider a sphere settling in still water ( $\rho_f = 1000 \text{ kg/m}^3$ , viscosity  $\nu = 10^{-6} \text{ m}^2/\text{s}$ ) at the same value of the particle Reynolds number  $Re'_p \equiv \omega_s d'_p/\nu$  (with  $\nu$  the kinematic viscosity) as in the simulations by Zhang et al. (2025),  $Re'_p \approx 914$  (using  $d'_p = d_p - 2l$ ). For this condition, flow separation occurs at a polar angle of about  $\theta = 80^\circ$  (Schlichting and Gersten, 2017), where  $\theta = 0$  corresponds to the bottom-most point of the settling sphere. Then, from analogy to the boundary layer development on a flat plate (Schlichting and Gersten, 2017), one obtains

$$\delta_{\text{max}} \approx 5\sqrt{\frac{\nu\theta d_p'/2}{1.5\omega_s \sin \theta}} \approx 0.11d_p'$$
 (7)

as an upper limit for  $\delta$  at the surface of the sphere, in which  $1.5\omega_s \sin\theta$  is an estimation of the outer flow velocity from potential flow approximation.

For the naturally-shaped grains ( $d_p=3.9~\mathrm{mm}$ ) and two shrinkage coefficients  $\mathrm{Sk}=[0.55,0.7]$  tested by Zhang et al. (2025), the corresponding slip lengths  $l=\mathrm{Sk}\Delta x=[0.275,0.35]~\mathrm{mm}$  are comparable to  $\delta_{\mathrm{max}}\approx[0.381,0.364]~\mathrm{mm}$  calculated from equation (7) using  $d_p'\approx d_p-2l=[0.86,0.82]d_p$  and  $\omega_s=[0.2726,0.2858]~\mathrm{m/s}$ . However, l should actually by much smaller than  $\delta(\boldsymbol{x}_s)$  in order for the first-order Taylor approximation to make sense. Furthermore, for surface points  $\boldsymbol{x}_s$  sufficiently away from the flow separation points, at sufficiently lower polar angles  $\theta$ ,  $\delta$  will even be much smaller than l. In summary, the fact that  $\delta$  is of comparable size down to much smaller than l falsifies the physical reasoning behind approximating Navier-slip conditions with the artificial-shrinkage method by Zhang et al. (2025).

## 2.2 Falsification with independent DNS-DEM simulations

We conducted independent DNS-DEM simulations of a sphere of diameter  $d_p = 4.2$  mm settling in a fluid at rest for two conditions:  $\rho_s = 2500 \text{ kg/m}^3$ ,  $\rho_f = 1000 \text{ kg/m}^3$ , and  $\rho_f \nu = 0.8 \text{ Pa.s}$  (condition 1) and  $\rho_s = 2471 \text{ kg/m}^3$ ,  $\rho_f = 998.23 \text{ kg/m}^3$ , and  $\rho_f \nu = 1.002 \times 10^{-3} \text{ Pa.s}$  (condition 2). Condition 1 is close to Stokes flow, whereas condition 2 is very similar to those studied by Deal et al. (2023) and Zhang et al. (2025). The simulations are based on the commercial code COMSOL Multiphysics® (COMSOL AB, 2024a), which contains modules for Navier-slip conditions. The mesh grid size  $\Delta x$  is recommended to be larger than the slip length l for the simulations to work well (COMSOL AB, 2024b). At the same time, the mesh must be sufficiently fine to resolve the salient features of the flow around the sphere. We found that  $\Delta x = d_p/16$  is a good compromise in that regard, since this value corresponds to about the coarsest mesh that still reproduces the expected behaviors of the settling drag coefficient  $C_{D_{\text{settle}}}$  for spheres in situations where these are known from previous studies, as shown below.

Figure 1 shows that, for condition 1, the "measured" settling drag coefficient  $C_{D_{\rm settle}}$  obtained from equation (4) approximately obeys the behavior previously determined by Feng (2010) for particle Reynolds numbers  $Re_p \equiv \omega_s d_p/\nu \leq 150$ , with deviations of less than 4% for  $l 

155

160

165

Figure 1. The "measured" settling drag coefficient  $C_{D_{\text{settle}}}$  obtained from equation (4) approximately captures the expected behavior of  $C_{D_{\text{settle}}}$  predicted by equation (8) after Feng (2010). In both equations,  $C_{D_{\text{settle}}}$  is calculated using the settling velocities  $\omega_s$  determined from DNS-DEM simulations of a settling sphere for condition 1 (close to Stokes flow) and various slip lengths l. Note that, to work well, l should be smaller than the mesh grid size  $\Delta x$ , which explains the slightly increasing deviation from the expected value with increasing l.

## 3 Alternative explanation of Zhang et al.'s data with a null hypothesis model

From the previous section, we conclude that the artificial-shrinkage method by Zhang et al. (2025) does not approximate Navier-slip conditions and that simulations based on this method therefore do not correspond to physically realistic scenarios. In this section, to support this conclusion, we first show that such simulations essentially solve the fluid equations of motion and corresponding fluid-grain interaction forces of a system with transformed values of  $\rho_s$ , g, S, and  $d_p$ , whereas grain-grain contact interactions are still based on the non-transformed variables (§3.1). For the settling of a single grain, where grain-grain interactions are absent, this system is physically meaningful, whereas for bedload transport, this system is also physically unrealistic. However, a simple, straightforward argument based on the geometry of the contact network between sedimentary grains is then used to argue that this unrealistic system is essentially equivalent to a physically realistic system (§3.2). We show that this realistic system predicts the very same dependence on the settling velocity  $\omega_s$  (which increases with shrinkage) as equation (1), but without invoking a grain shape correction.

Henceforth, we introduce new notation: a quantity with a prime shall indicate its general value, whereas a quantity without a prime shall indicate its value for simulations with Sk = 0, termed non-shrunk simulations; for example,  $\omega_s = \omega_s'|_{Sk=0}$ . This notation is consistent with the previous definition of the volume-equivalent diameter  $d_p'$  of the virtual, shrunk grain. However,

Figure 2. Settling drag coefficient  $C_{D_{\text{settle}}}$  obtained from equation (4) for condition 2, using the settling velocities  $\omega_s$  determined from DNS-DEM simulations of a settling sphere, versus nondimensionalized slip length  $l/d_p$ . The blue symbols correspond to actual Navier-slip boundary conditions on the sphere surface, the red symbols to the approximation from the artificial-shrinkage method by Zhang et al. (2025), where a no-slip condition is applied to the surface of a virtual shrunk sphere with diameter  $d_p' = d_p - 2l$ . Note that the values of  $C_{D_{\text{settle}}}$  for the open blue symbols are less reliable and should be treated with caution, since the slip length l exceeds the mesh grid size  $\Delta x$ .

equation (6) now changes to

$$C'_{D_{\text{settle}}} = \frac{4(\rho_s/\rho_f - 1)gd_p^3}{3\omega_s'^2 d_p'^2} = \frac{\omega_s^2 d_p^2}{\omega_s'^2 d_p'^2} C_{D_{\text{settle}}},\tag{9}$$

since the meaning of  $\omega_s$  has changed. Furthermore, when using this notation and limiting our considerations to the varying-shrinkage simulations (Sk = [0,0.55,0.7]) by Zhang et al. (2025), which apart from the value of Sk are otherwise nearly identical to each other ( $\mu^* \approx \text{const}$  due to only very slight variation in S), the grain-shape-corrected bedload model by Deal et al. (2023), equation (1), essentially condenses to the functional form

$$\frac{q}{d_p \sqrt{(\rho_s/\rho_f - 1)gd_p}} = f\left(\frac{\omega_s^2}{\omega_s'^2} \frac{\tau_b}{(\rho_s - \rho_f)gd_p}\right),\tag{10}$$

where f denotes the same power-3/2 law as in equation (1), but with a modified prefactor. This is the relationship that will be derived in §3.2, but without invoking a grain shape correction.

## 3.1 Artificial-shrinkage method in transformed variables

In this section, we show that the simulations by Zhang et al. (2025) based on their artificial-shrinkage method can be reinterpreted in a meaningful manner using transformed values of  $\rho_s$ , g, S, and  $d_p$ . To demonstrate this, we first discuss the case of zero shrinkage, Sk = 0.

#### 180 3.1.1 Non-shrunk grains

In the reference case of non-shrunk grains, the LES solver numerically solves the following fluid momentum balance (Zhang et al., 2025):

$$\rho_f D_t \mathbf{u}_f = \nabla \cdot \boldsymbol{\sigma}_q + \rho_f g_x \hat{\mathbf{x}},\tag{11}$$

where  $D_t \equiv \partial_t + \boldsymbol{u}_f \cdot \boldsymbol{\nabla}$  denotes the material derivative,  $\boldsymbol{u}_f$  is the flow velocity, and  $\boldsymbol{\sigma}_g$  the stress tensor, with the subscript 'g' indicating that the vertical component of the gravitational body force term  $\rho_f g_z \hat{\boldsymbol{z}}$  has been lumped into the fluid pressure. Based on the solution of this equation, the total force  $\boldsymbol{F}_p$  on a grain p is then calculated as

$$\boldsymbol{F}_{p} = \int_{\mathcal{S}_{p}} \boldsymbol{n}_{p} \cdot \boldsymbol{\sigma}_{g} dS - \rho_{f} V_{p} g_{z} \hat{\boldsymbol{z}} + \rho_{s} V_{p} \boldsymbol{g} + \boldsymbol{F}_{p}^{c}, \tag{12}$$

where  $S_p$  denotes the surface of p and  $n_p$  the outward-directed normal vector on it,  $\rho_s$  is the grain density,  $V_p = \pi d_p^3/6$  the grain volume, and  $F_p^c$  the contact force acting on p. In equation (12), the first term on the right-hand side represents the non-buoyancy fluid-grain interaction force and the second term the buoyancy force. It is important to be aware that equations (11) and (12) are a mathematically equivalent simplification of the actual physical equations

$$\rho_f D_t \boldsymbol{u}_f = \boldsymbol{\nabla} \cdot \boldsymbol{\sigma} + \rho_f \boldsymbol{g}, \tag{13}$$

$$\boldsymbol{F}_{p} = \int_{\mathcal{S}_{p}} \boldsymbol{n}_{p} \cdot \boldsymbol{\sigma} dS + \rho_{s} \boldsymbol{g} V_{p} + \boldsymbol{F}_{p}^{c}, \tag{14}$$

in which  $\sigma$  is the actual physical fluid stress tensor responsible for the total fluid-grain interaction (non-buoyancy and buoyancy). It is related to  $\sigma_q$  through

$$\boldsymbol{\sigma} = \boldsymbol{\sigma}_g - \rho_f g_z z \boldsymbol{I},\tag{15}$$

with I the identity tensor. That is, by solving equations (11) and (12), one actually solves the physical equations (13) and (14).

## 3.1.2 Shrunk grains

190

Zhang et al. (2025) state that, in the case of shrunk grains, the LBM solver sees a smaller grain volume  $V_p' = \pi d_p'^3/6$ , but the DEM solver still sees the non-shrunk grain volume  $V_p$ . They further state that the buoyancy and gravitational forces are

calculated based on  $V_p$  rather than  $V_p'$ . In mathematical terms, this means they solve the following equations:

$$\rho_f D_t \boldsymbol{u}_f = \boldsymbol{\nabla} \cdot \boldsymbol{\sigma}_{q'} + \rho_f g_x \hat{\boldsymbol{x}}, \tag{16}$$

$$\boldsymbol{F}_{p} = \int_{\mathcal{S}_{p}'} \boldsymbol{n}_{p} \cdot \boldsymbol{\sigma}_{g'} dS - \rho_{f} V_{p} g_{z} \hat{\boldsymbol{z}} + \rho_{s} V_{p} \boldsymbol{g} + \boldsymbol{F}_{p}^{c}, \tag{17}$$

where the prime indicates quantities associated with the smaller grain volume  $V_p'$  (consistent with the earlier definition of primed quantities). The question is now, what are the actual physical equations that are being solved through solving equations (16) and (17)? In other words, what are the analogs to equations (13) and (14)? It can be shown that these are the following equations

$$\rho_f D_t \boldsymbol{u}_f = \boldsymbol{\nabla} \cdot \boldsymbol{\sigma'} + \rho_f \boldsymbol{g'}, \tag{18}$$

$$\boldsymbol{F}_{p} = \int_{\mathcal{S}_{p}'} \boldsymbol{n}_{p} \cdot \boldsymbol{\sigma}' dS + \rho_{s}' \boldsymbol{g}' V_{p}' + \boldsymbol{F}_{p}^{c}, \tag{19}$$

210 in which

$$\rho_s' \equiv (V_p/V_p')\rho_s,\tag{20}$$

$$\mathbf{g}' \equiv (S'g', 0, -g'),\tag{21}$$

$$S' \equiv \left(\frac{1 - \rho_f/\rho_s'}{1 - \rho_f/\rho_s}\right) S,\tag{22}$$

$$g' \equiv \left(\frac{1 - \rho_f/\rho_s}{1 - \rho_f/\rho_s'}\right) g,\tag{23}$$

215 
$$\sigma' \equiv \sigma_{g'} - \rho_f g_z' z I$$
 (24)

are the transformed variables. This can be readily confirmed through substituting equations (20–24) into equations (18) and (19). Equations (18) and (19) are, in terms of mathematical structure, equivalent to equations (13) and (14). This means, in their shrunk-grain simulations, Zhang et al. (2025) effectively solve a physical system in which grains have an increased density  $\rho'_s$  and a decreased volume  $V'_p$  (but their mass  $\rho'_s V'_p = \rho_s V_p$  remains unchanged), while the bed slope exhibits the larger value S' and the magnitude of the vertical component of the gravitational acceleration the smaller value g'.

For the settling of a single grain, where grain-grain interactions are absent ( $\mathbf{F}_p^c = 0$ ), the transformed system above is physically meaningful. In particular, as required, the associated settling drag coefficient  $C'_{D_{\text{settle}}}$  is the same in transformed and non-transformed variables, consistent with equation (9):

$$C'_{D_{\text{settle}}} = \frac{4(\rho'_s/\rho_f - 1)g'd'_p}{3\omega'^2_s} = \frac{\omega_s^2 d_p^2}{\omega'^2_s d'^2_p} C_{D_{\text{settle}}}.$$
 (25)

However, in the case of bedload transport, where  $F_p^c \neq 0$ , the contact force  $F_p^c$  is still calculated under the assumption that grains have the non-shrunk volume  $V_p$  rather than  $V_p'$ , which means the simulated system is still unphysical. This problem is addressed below.

## 3.2 Bedload flux for shrunk-grain simulations from contact geometry similarity

Since the grains' contact dynamics calculated by the DEM solver depends on only the contact geometry (e.g., ratio of contacting grain sizes), one expects that the behavior of the unphysical transformed system is essentially equivalent to that of a physical system in which all grains are shrunk by the same ratio from  $d_p$  to  $d'_p$  also from the point of view of the DEM solver:

$$q'(\tau_b', \rho_s', \rho_f, g', S', d_p', \mathbf{F}_p^c) \approx q'(\tau_b', \rho_s', \rho_f, g', S', d_p', \mathbf{F}_p^{c'}), \tag{26}$$

where  $F_p^{c\prime}$  is the corresponding transformed contact force. The null hypothesis is that this physical system can be explained by a classical functional relationship of the form

235 
$$\frac{q'}{d'_p \sqrt{(\rho'_s/\rho_f - 1)g'd'_p}} = f\left(\frac{\tau'_b}{(\rho'_s - \rho_f)g'd'_p}\right),\tag{27}$$

like equation (1) by Deal et al. (2023), but without its grain-shape-parametrizing modification by  $C^*/\mu^*$ .

To evaluate the consequences of this null hypothesis, we need to understand how  $\tau_b$  and q transform to  $\tau_b'$  and q', respectively. First, using equations (22) and (23), we obtain

$$\tau_b' = \rho_f S' g' h = \rho_f S g h = \tau_b, \tag{28}$$

240 where h is the water depth. Second, we employ the classical partition of q' into the sediment load  $\chi'$  and the average sediment transport velocity v' (Bagnold, 1956),

$$q' = \chi' v', \tag{29}$$

to derive the transformation of q. The sediment load  $\chi'$  can be obtained from integrating the particle volume fraction  $\phi$  from the bed surface elevation, z=0, to the top of the bedload layer,  $z=h_b'$  (Bagnold, 1956):

245 
$$\chi' = \int_{0}^{h_b'} \phi dz = \overline{\phi} h_b'. \tag{30}$$

Since the bedload layer thickness  $h_b'$  scales with  $d_p'$ ,  $\chi'$  transforms as

$$\chi' = \frac{d_p'}{d_p} \chi. \tag{31}$$

Furthermore, the appropriate scale for the sediment transport velocity v' is the settling velocity  $\omega'_s$  (Bagnold, 1956). Hence, v' transforms as

$$250 \quad v' = \frac{\omega_s'}{\omega_s} v. \tag{32}$$

Using equation (9), equations (31) and (32) lead to

$$q' = \sqrt{\frac{C_{D_{\text{settle}}}}{C'_{D_{\text{cettle}}}}} q. \tag{33}$$

**Figure 3.** Test of null hypothesis model, equation (34), against varying-shrinkage simulation data by Zhang et al. (2025) for natural gravel (NG) grains in a narrow-flume ("flume") or wide-channel ("wide") configuration. The symbol code is the same as in their figures 7(c) and 7(d). Those of the symbols of their figures 7(c) and 7(d) that do not appear in the present plot are from experiments or spherical grain (SP) conditions for which no shrunk-grain simulations were carried out.

Finally, inserting the transformations equations (20), (23), and (28) into equation (27) yields

$$\sqrt{\frac{C_{D_{\text{settle}}}}{C'_{D_{\text{settle}}}}} \frac{q}{d_p \sqrt{(\rho_s/\rho_f - 1)gd_p}} = f\left(\frac{C_{D_{\text{settle}}}}{C'_{D_{\text{settle}}}} \frac{\omega_s^2}{\omega_s'^2} \frac{\tau_b}{(\rho_s - \rho_f)gd_p}\right). \tag{34}$$

Equations (34) is equivalent to equation (10), the condensed version of equation (1) by Deal et al. (2023), except for additional rescalings of both sides by a power of  $C_{D_{\rm settle}}/C'_{D_{\rm settle}} = C'_{D_{\rm settle}}|_{\rm Sk=0}/C'_{D_{\rm settle}}$ . This drag coefficient ratio is nearly equal to unity for the conditions studied by Zhang et al. (2025), where the dependence of  $C'_{D_{\rm settle}}$  on the shrunk-grain particle Reynolds number  $Re'_p$  is very weak. (This is also evident from the fact that the solid line in figure 2 captures the trend of the red symbols.) In fact, the rescaling in equation (34) collapses the varying-shrinkage simulations by Zhang et al. (2025), as shown in figure 3.

#### 4 Conclusions

265

We have shown that a recently introduced numerical method to independently vary the fluid-particle interaction force experienced by transported grains in grain-resolved bedload transport simulations is unphysical. The method in question was proposed by Zhang et al. (2025) and consists of artificially shifting the locations of the no-slip boundary conditions from the actual grain surface to a virtual surface a distance l into the grain interior. These authors hoped that this method would well approximate Navier-slip conditions with a slip length l for hydrodynamic conditions that are typical for turbulent bedload transport. However, our analytical and numerical analyses clearly falsify this hypothesis (§2), implying that their method does not correspond to physically realistic scenarios.

Zhang et al. (2025) introduced their method as a simple means to test the bedload transport model by Deal et al. (2023), to date the probably only bedload transport model that attempts to account for the typically non-spherical shape of transported grains. The problem was that the grain-shape-parametrizing coefficient in this model,  $C^*/\mu^*$  in equation (1), varied by less than 20%, only between 0.84 and 1.05, in the original experiments by Deal et al. (2023), with four of their five tested grain materials even exhibiting nearly no variation at all  $(C^*/\mu^* \in [1.01, 1.05])$ . However, the newly generated numerical data by Zhang et al. (2025) do not alleviate this shortcoming due to the falsification of their method. This is further supported by the fact that an alternative bedload transport model that does not invoke grain shape corrections is also able to capture these data (§3). Hence, the question of how to properly account for grain shape variations in bedload transport remains an unresolved problem.

*Code and data availability.* The code used to produce figures 1 and 2 is commercially available (COMSOL AB, 2024a). The data in figures 1 and 2 will be available in an online repository. For the data and code in and behind figure 3, see Zhang et al. (2025).

Author contributions. Y.C. conducted the numerical simulations. Y.C., T.P. analyzed the data. T.P. derived the boundary layer thickness estimation. O.D., T.P. derived the null hypothesis bedload transport model. T.P., Y.C. wrote the paper. All authors discussed the results and implications and commented on the paper at all stages.

Competing interests. O.D. is a member of the editorial board of Earth Surface Dynamics.

Acknowledgements. T.P. acknowledges support from grants National Natural Science Foundation of China (12350710176, 12272344). O.D. acknowledges support from Texas A&M Engineering Experiment Station.

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
