# Peer review of "On the testing of grain shape corrections to bedload transport equations with grain-resolved numerical simulations"

_EGUsphere, 2025_

## Author Comment (AC1)

**Response to Reviewer 1**

**1.** *This manuscript examines the validity of an "artificial-shrinkage" method used in grain-resolved bedload transport simulations to mimic Navier-slip boundary conditions. The authors combine analytical boundary-layer estimates with independent DNS-DEM settling simulations to show that the imposed slip lengths are not small relative to the boundary-layer thickness, meaning the method cannot reproduce true Navier-slip behavior. Their results demonstrate that the resulting drag coefficients are substantially overestimated and do not correspond to any physically realistic fluid–particle interaction. They further present a null-hypothesis model showing that the trends in Zhang et al. (2025) can be reproduced without invoking grain-shape corrections, but simply by accounting for the virtual grain size used in the artificial shrinkage. Overall, the manuscript argues convincingly that the artificial-shrinkage method is inappropriate for testing grain-shape corrections in bedload transport equations. This reviewer has nothing to add to the arguments provided here.*

We thank the reviewer for this positive assessment of our manuscript.

**Response to Reviewer 2**

**1.** *Deal et al. investigated the effect of grain shape on sediment transport. To address certain limitations in their experimental setup, Zhang et al. conducted numerical simulations, which largely corroborated the findings of Deal et al. In the present manuscript, the Authors aim to challenge the methodology of Zhang et al. and, consequently, the conclusions of Deal et al. Their argument is structured around three components: (1) an analytical critique, (2) a critique based on independent DNS–DEM simulations, and (3) an alternative interpretation of the results presented by Zhang et al.*

*I would like to congratulate the authors on their work, as the three arguments they present are particularly convincing. First, they point out that the shrinking length in Zhang et al. is comparable to the boundary-layer thickness, even though it should be much smaller. Second, based on independent simulations, they show that for a given $l/d$, imposing a Navier slip condition at $l$ and a no-slip condition on a smaller sphere produces different results. Third, they offer an alternative explanation for the findings of Zhang et al.*

We thank the reviewer for this positive assessment of our manuscript.

**2.** *I must confess that I found the manuscript difficult to follow, primarily because the overall structure of the argument is unclear. The paper does not follow the conventional Introduction–Methods–Results–Discussion format typically expected in this field. The Introduction is unusually dense, and in my opinion, the extensive use of equations at this stage is not ideal.*

*A dedicated Discussion section is also missing. Instead, Methods and Results are interwoven and organized around the three proposed "falsification" steps. I would encourage the Authors to reorganize the manuscript for greater clarity. For instance, the numerical simulation description (currently in Section 2.2) and the derivation of equations in transformed and non-transformed coordinates (Sections 3.1.1 and 3.1.2) would be more appropriately placed in a consolidated Methods section. The Results section could then present the core findings (e.g., Figures 1 and 2), followed by a Discussion that interprets these findings in light of the analytical and numerical critiques (Sections 2.1 and 3.2, Figure 3). Of course, different reorganizations are possible according to the authors' personal taste.*

In the Introduction, we want to introduce Zhang et al.'s (2025) artificial shrinkage method. In order to do this, we need to explain why these authors invented that method in the first place, which means we need to introduce the grain-shape-parametrizing bedload transport model by Deal et al. (2023). The easiest way to introduce this model is by showing its equations. An unambiguous introduction of the problem without equations would be very difficult. Whatever we were to do, there would be a substantial chance that the content is being misunderstood.

Following the conventional Introduction–Methods–Results–Discussion format for our paper has significant drawbacks, which is why we decided against it. For example, the reviewer suggested moving the derivation of the equations Sections 3.1.1 and 3.1.2 to the Methods. However, in that case, one would still need to explain in the Results (or Discussion if Section 3.2 were moved to there) what exactly the non-transformed and transformed coordinates are, since this is needed to understand almost all the equations of Section 3.2. And the only way in which one can explain what these coordinates are is by presenting the equations for the transformed variables, Eqs. (18)-(24). In our opinion, there is no way around that. Of course, one might move only Section 3.1.1 to the Methods, but we think the back-to-back comparison between Section 3.1.1 and Section 3.1.2 is very instructive and beneficial for readability. Concerning the numerical description in Section 2.2, the actual description of the numerical method in Section 2.2 is merely four sentences long, we do not think that this justifies having a separate Methods section either. And concerning a Discussion/Summary section, it would be very short. A brief overall summary is already contained in the Conclusions.

In order to address the readability critique of the reviewer, we added a paragraph to the end of the introduction, explaining why the manuscript does not have a separate Methods section and mentioning that Sections 2 and 3 are self-contained (i.e., they contain the respective Methods, Results, and Summaries). To make them indeed self-contained, we added one-sentence summaries to the ends of Sections 2.1, 2.2, and 3, respectively.

**3.** *That being said, I recommend that the present manuscript be rejected with major revisions, not because the scientific arguments are unconvincing (indeed, they are sound and*

*compelling) but because the current structure significantly impedes readability. Clear and accessible presentation of results is, in my view, essential for Earth Surface Dynamics. Nevertheless, I leave the final decision to the Associate Editor: if the AE considers the current organization to be compatible with the journal's editorial standards, then acceptance in its present form would, of course, be appropriate.*

We would like to emphasize that the current structure of the manuscript is consistent with the guidelines of Earth Surface Dynamics: `https://www.earth-surface-dynamics.net/submission.html`. There seems to be no obligation to have separate Methods and Discussion/Summary sections. We therefore hope that the Associate Editor will not ask us to reorganize the manuscript.